# SurgicAI: A Hierarchical Platform for Fine-Grained Surgical Policy Learning and Benchmarking

Jin Wu[1]    Haoying Zhou[2]    Peter Kazanzides[1]    Adnan Munawar[1]    Anqi Liu[1]

[1]Johns Hopkins University, Baltimore    [2]Worcester Polytechnic Institute, Worcester
{jwu220, pkaz, amunawa2, aliu.cs}@jhu.edu    hzhou6@wpi.edu

## Abstract

Despite advancements in robotic-assisted surgery, automating complex tasks like suturing remains challenging due to the need for adaptability and precision. Learning-based approaches, particularly reinforcement learning (RL) and imitation learning (IL), require realistic simulation environments for efficient data collection. However, current platforms often include only relatively simple, non-dexterous manipulations and lack the flexibility required for effective learning and generalization. We introduce SurgicAI, a novel platform for development and benchmarking that addresses these challenges by providing the flexibility to accommodate both modular subtasks and more importantly task decomposition in RL-based surgical robotics. Compatible with the da Vinci Surgical System, SurgicAI offers a standardized pipeline for collecting and utilizing expert demonstrations. It supports the deployment of multiple RL and IL approaches, and the training of both singular and compositional subtasks in suturing scenarios, featuring high dexterity and modularization. Meanwhile, SurgicAI sets clear metrics and benchmarks for the assessment of learned policies. We implemented and evaluated multiple RL and IL algorithms on SurgicAI. Our detailed benchmark analysis underscores SurgicAI's potential to advance policy learning in surgical robotics. Details: https://github.com/surgical-robotics-ai/SurgicAI

## 1 Introduction

Robotic-assisted surgery (RAS) has revolutionized the medical field by enhancing precision, reducing surgeon fatigue, and significantly shortening recovery times. While existing RAS systems lack automation, arguably automation has the potential to improve upon the current standard of care and surgeon workload. However, automating complex surgical tasks such as suturing remains a significant challenge. Traditional methods [1–5] rely on pre-programmed instructions and heuristics, lacking the adaptability and generality needed for diverse surgical scenarios. Additionally, there is a lack of fine-grained platforms for complex, multi-stage tasks that provide rich data and assessment metrics. These limitations highlight the need for advanced learning frameworks to handle the complexities of surgical manipulation, as well as comprehensive evaluation and benchmarking methods.

Reinforcement learning (RL) and imitation learning (IL) have emerged as powerful mechanisms in robotic automation, exhibiting the potential to transform the field of surgical robotics. By enabling robots to learn from interactions with their environment and from expert demonstrations, these learning paradigms promise to significantly enhance the autonomy and effectiveness of surgical robots. Recent research [6–8] has focused on developing simulation environments for implementing machine learning algorithms for various tasks. However, previous work on surgical application often compromises on either photorealism [9, 10] or physical realism [11]. Additionally, these platforms typically involve only relatively simple surgical tasks.

38th Conference on Neural Information Processing Systems (NeurIPS 2024) Track on Datasets and Benchmarks.

In this paper, we introduce SurgicAI, which features complex deformable thread simulation, and real-time interaction compatible with the da Vinci Research Kit (dVRK) [12], providing a comprehensive learning environment for developing and testing robotic suturing techniques. Moreover, SurgicAI integrates standardized pipelines, advanced hierarchical learning frameworks, diverse task suites, and benchmark performance metrics to evaluate its capabilities.

The primary contributions of SurgicAI are the following:

- **Standardized Pipeline:** SurgicAI offers a comprehensive and standardized pipeline for the entire process of surgical automation. This pipeline includes collecting and preprocessing expert trajectories from teleoperation devices, training agents using reinforcement learning and imitation learning approaches, and evaluating their performance.

- **Hierarchical Task Decomposition:** Most platforms mentioned previously [9–11] struggle with relatively complex, multi-stage tasks like suturing. SurgicAI is the first platform to implement an open-source hierarchical learning framework specifically for managing multi-stage surgical procedures. The hierarchical structure facilitates the decomposition of intricate surgical tasks into smaller, more manageable subtasks, each handled by specialized modules. This approach not only enhances learning efficiency but also promotes the reuse of learned skills across different procedures, thereby improving the generality and scalability of robotic systems. Figure 1 shows how we can train both high-level policy and low-level policies to conduct the multi-stage procedures.

- **Diverse Suite of Tasks:** SurgicAI encompasses a wide variety of manipulative operations required during the suturing, such as approaching and grasping the needle, inserting the needle, and regrasping the needle. Each of these tasks demands a high level of dexterity and precision, contributing significantly to the automation of suturing processes.

- **Benchmark of Performance:** SurgicAI establishes clear metrics for each suturing task and benchmarks performance across various RL and IL algorithms. With support for the Gymnasium API [13], and RL libraries like Stable Baseline3 (SB3) [14] and d3rlpy [15], SurgicAI offers a user-friendly interface for defining new tasks and implementing custom algorithms. With baselines and metrics, SurgicAI also enables researchers to compare and improve their methods.

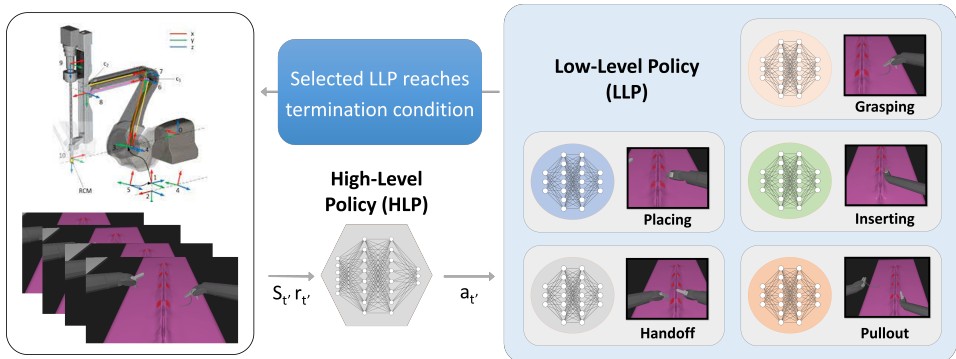

Figure 1: Hierarchical Framework in SurgicAI. A High-Level Policy (HLP) selects and coordinates Low-Level Policies (LLPs) for specific tasks like grasping, placing, inserting, handoff, and pullout. Each LLP manages actions within its designated subtask until a termination condition is met, after which control returns to the HLP for the next decision.

## 2 Related Work

### 2.1 Simulation Platforms for Data Collection and Training

The development of simulation environments for data collection and training in surgical robotics has advanced significantly, driven by the need for scalable and efficient methods to train and evaluate algorithms. Frameworks such as dVRL [9] and SurRoL [10] integrate real-time physics engines with user-friendly RL libraries. Both are compatible with dVRK and have demonstrated successful policy transfer to real-world scenarios. Other work like Surgical Gym enhances data collection efficiency

through high-performance GPU-based simulations, achieving significantly faster sampling rates compared to previous platforms. However, it is exclusively designed for RL implementation and does not offer interfaces for teleoperation or collecting expert demonstrations for imitation learning or demonstration-guided purposes. Additionally, most simulations lack soft body simulation, except for recent developments in SurRoL [16]. This limits their physical realism.

Recent advancements, exemplified by ORBIT-Surgical [17], have introduced photorealistic and physics-based environments with GPU parallelization, supporting both imitation learning and reinforcement learning. Despite these improvements, such platforms are limited to relatively simple tasks like reaching and placing, and they struggle with more complex, multi-stage tasks such as suturing. In contrast, the AccelNet Surgical Robotics Challenge (SRC) [18] offers a comprehensive solution by addressing these limitations and providing robust support for complex suturing procedures. Our work provides the necessary interface and library support to train and test RL and IL algorithms with low-level dynamics provided in SRC, using a hierarchical framework for long-term planning.

## 2.2 Learning Based Approaches in Surgical Automation

Robot learning has been applied to various surgical scenarios, including debris removal [19–21], tissue retraction [22–24], suturing [25–27], and cutting [28, 29]. The results from these applications show promising potential for machine learning-based approaches to become key solutions for surgical automation. However, traditional RL methods face significant challenges in these contexts due to sparse rewards, discounting issues, and exploration difficulties inherent in high-dimensional and long-horizon tasks [30, 31]. These challenges often hinder RL algorithms from effectively learning and performing complex surgical tasks. To address these issues, many existing methods rely heavily on elaborate reward shaping to facilitate extensive exploration [22, 32]. This process typically involves subjective manual engineering, which can introduce biases and lead to unexpected policies that get stuck in local optima [31]. The reliance on handcrafted rewards can also limit the generality of the learned policies, as they may not perform well in slightly different or unforeseen scenarios.

Advanced methodologies, such as [33, 34], have been developed to enhance exploration efficiency by incorporating demonstration data into RL. This approach allows learning from expert knowledge, reducing the time and computational resources needed to achieve effective policy learning. Work from [35, 36, 27] predefined primitive skills and sequences them at transition points, creating a more structured and guided learning process. These methods help break down complex tasks into simpler, more manageable components, which can be learned more effectively.

Building on these advancements, the hierarchical framework [37–39], utilized in SurgicAI, has been introduced to better coordinate sub-policies. This hierarchical approach divides the overall task into a hierarchy of sub-tasks, each with its own specific policies. By structuring the learning process in this way, the hierarchical framework not only improves learning efficiency but also enhances the robustness and adaptability of the system. Moreover, SurgicAI distinctively pioneers the application of image-guided imitation learning algorithms for complex manipulative tasks involving patient-side manipulators (PSMs). While platforms such as [10] and [11] have defined some basic Endoscopic Camera Manipulator (ECM) tracking tasks using image data, and [17] demonstrated the capability for collecting RGB or segmented image data, they have not extensively applied imitation learning to the intricate and precise manipulations required for tasks like suturing.

## 3 Our Framework

### 3.1 Simulation Environment

The simulation environment of SurgicAI is based on the SRC [18] and utilizes the Asynchronous Multi-Body Framework (AMBF) [40, 41]. This environment includes two Patient Side Manipulators (PSMs) from the da Vinci Surgical System, an Endoscopic Camera Manipulator (ECM), a needle with or without thread, and realistic phantoms marked with entry and exit holes for suturing. A key feature of this environment is the simulation of a deformable thread, its interaction with the phantom, and the ability to pass it through the specified holes in the phantom, which significantly enhances the realism of tissue simulations. The system is fully compatible with the dVRK system, enabling real-time communication with various teleoperation devices such as dVRK MTMs, Geomagic, and Razer Hydra. Additionally, our environment supports a headless mode to facilitate efficient data sampling

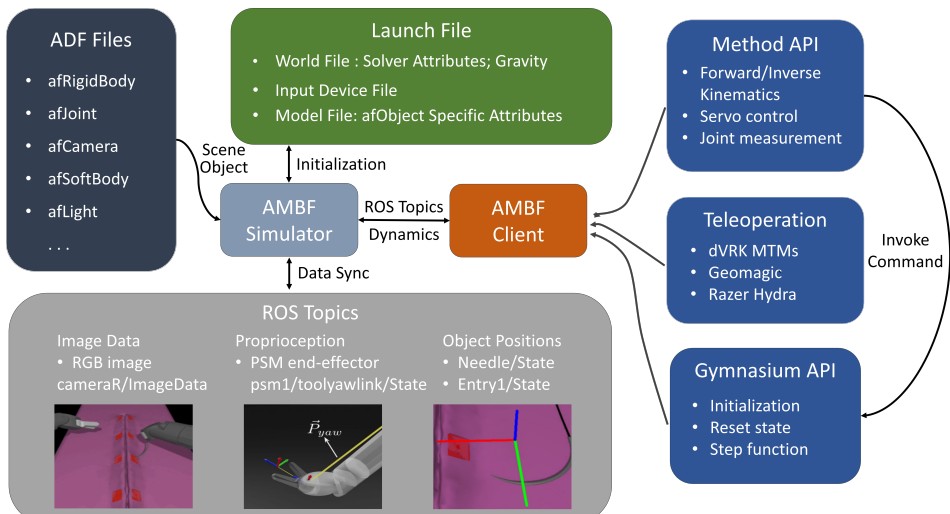

Figure 2: Detailed Workflow of our Simulation Environment. The AMBF Description Files (ADF) define various simulation objects, such as rigid bodies, joints, and cameras. The launch file specifies simulation parameters and model-specific details. The AMBF Simulator processes these parameters to initialize the environment, while the AMBF Client bridges the simulator and the control scripts via ROS topics. Method APIs provide essential functions for controlling the simulation, including kinematics and servo control, while teleoperation scripts enable connections with multiple devices. The Gymnasium API integrates RL and IL capabilities, optimizing policies through interaction with the control scripts.

without the computational overhead of graphical rendering. The system generates multi-modal data, including ground truth positions, RGB images, and labeled depth maps, making it a suitable platform for data-driven approaches to suturing automation. The workflow is detailed in Figure 2.

## 3.2 Environment Settings

Our simulation environment adheres to the Gymnasium framework rules [13], ensuring compatibility with various RL and IL tasks.

**Observation and State Space:** This environment features a versatile observation space that accommodates both low-dimensional and high-dimensional states for comprehensive data collection. In low-dimensional settings, the observation space includes ground-truth 6D poses of objects such as needles, entry and exit holes, and end-effectors. When using Hindsight Experience Replay (HER) [42], the agent receives extra lists of achieved and desired goals, facilitating learning from past experiences. High-dimensional settings, on the other hand, comprise scene-invariant RGB images from multiple views and proprioception data from PSMs, specifically the 6D pose of the end-effector in its local frame. Our settings support the online and offline training for both low and high dimensional state space. In our experiments, we utilize low-dimensional states in online and offline RL for rapid validation, whereas high-dimensional states are reserved for offline imitation learning, as it allows us to bypass the time-intensive process of collecting online image data.

**Action Space and Reward Setting:** For low-level policies, the action space is continuous and features 7 DOF vector $a \in \mathbb{R}^7$. The agent controls the x, y, z, roll, pitch, and yaw of the end-effector in its local frame, along with the state of the jaw (0 for closed, 1 for open). These action steps are restricted in ($\delta_x$, $\delta_y$, $\delta_y$, $\delta_{roll}$, $\delta_{pitch}$, $\delta_{yaw}$, $\delta_{jaw}$). The terms represent the action step sizes or incremental changes allowed in each corresponding direction during each action. Additionally, to ensure that the PSMs do not exceed their physical limits, their positions are constrained within predefined boundaries. To mitigate distribution shift, we ensure that the initial state of each subtask encompasses the potential terminal state of the preceding task while permitting additional exploration. In terms of high-level policy, the action space is discrete, with actions represented by integral numbers ranging from 0 to 4, each corresponding to the task number to be executed. All the error terms are calculated using the L2 norm, and the reward functions are defined in two ways: sparse reward, which

is 0 if successful and -1 otherwise, and dense reward, given by $-(d_{\text{trans}}/100 + d_{\text{angle}}/10)$, where $d_{\text{trans}}$ and $d_{\text{angle}}$ represent the translation and orientation errors, respectively.

## 3.3 Task Overview

To address the complexity of the suturing process, we have segmented the process into several distinct subtasks, similar to [36]: approaching and grasping the needle, placing the needle at the entry hole, inserting the needle, handing off the needle (regrasping it with the other PSM), and pulling out the needle. This segmentation is informed by empirical observations that show users tend to slow down during transitions between these tasks, making it easier to define and separate each subtask. For specific low-dimensional settings, each subtask is defined as below. While the settings described below offer functionally plausible thresholds for successful task execution, the numerical thresholds for distances and orientation errors are also flexible to adapt to different levels of precision.

**Grasping:** At the beginning of each episode, the needle's position is randomized. A specific point on the needle is designated as the grasping point. The episode is considered successful if the remaining distance between the jaw and the grasping point is within 1 mm, the orientation error is within 10 degrees, and the contact sensor attached to the gripper successfully detects the needle object.

**Placing:** The initial state involves the jaw randomly grasping the needle with positional and orientation errors of up to 1.5 mm and 12 degrees respectively from the expected grasping orientation. Eventually, the needle tip should align with the entry point's center and perpendicular to the entry plane. Success criteria are a distance within 5 mm and an orientation error within 10 degrees.

**Inserting:** The initial state includes the same randomized grasping as in the placing task, with the needle tip positioned within 5.5 mm and 12 degrees of the entry point. The goal is for one-third of the needle to remain outside the exit point, with the tangential direction properly aligned. Success criteria are a remaining distance within 5 mm, an orientation error within 10 degrees, and the needle passing through both entry and exit holes.

**Handoff:** Starting with the needle passing through the phantom, a grasping point on the needle is identified. Success is achieved if the remaining distance between the jaw and the grasping point is within 2 mm, the orientation error is within 15 degrees, and the contact sensor detects the needle.

**Pullout:** The initial state involves the jaw randomly grasping the needle with a positional error of up to 1.5 mm and an orientation deviation up to 20 degrees from the expected grasping point in the handoff task. The goal is to fully extract the needle from the phantom and reach a predetermined end point. Success criteria are a distance within 5 mm and an orientation error within 20 degrees.

This modular approach offers significant advantages in terms of flexibility and reusability. Each subtask can be fine-tuned independently to adapt to different scenarios. For instance, the insertion policy can be modified to accommodate changes in the geometry of the phantoms while other policies remain unchanged. The grasping policy can be adjusted for various types of grasping actions. By organizing and integrating these pre-trained subtasks, the agent is able to perform the repetitive suturing procedure effectively.

## 3.4 Hierarchical Architecture for Multi-stage Tasks

The hierarchical architecture is designed to efficiently manage complex robotic tasks by decomposing them into smaller, manageable subtasks, each handled by specialized neural networks. This approach leverages both high-level and low-level policies to ensure precise control and coordination, leading to successful task execution.

**High-Level Policy (HLP):** At the core of the hierarchical architecture is the High-Level Policy, which utilizes a neural network for decision making. The HLP is responsible for sequencing and coordinating the Low-Level Policy. Based on the current state of the task, the HLP determines the appropriate subtask to activate, managing transitions between subtasks and ensuring that each stage of the task is executed in the correct order. This component allows the system to adapt to dynamic changes in the environment and varying initial configurations.

**Low-Level Policy (LLP):** The Low-Level Policies are specialized neural networks trained to execute specific subtasks. Each LLP is responsible for a particular aspect of the overall task, such as grasping, placing, inserting, handoff, or pullout. These subtasks are parameterized and trained independently,

enabling the LLPs to master their respective tasks efficiently. The specialization of LLPs enables focused learning and optimization, achieving more robust and reliable performance for each subtask.

**Integration and Coordination:** The hierarchical framework operates in a cyclical manner to ensure efficient task execution: 1) The HLP generates a subtask based on the current state of the environment; 2) The corresponding LLP executes the subtask and reaches its terminal state, upon which the process loops back to the HLP for the next subtask generation. This process continues iteratively until the overall task is completed. By breaking down complex tasks into simpler components, the hierarchical architecture ensures that each subtask is handled by a network specialized in that particular operation. The HLP's role in managing the sequence and coordination of subtasks is crucial for maintaining the overall coherence and efficiency of the task execution.

## 4 Capability of Our Framework

### 4.1 Platform Usage Guideline

Figure 3 illustrates the training pipeline for RL agents. The pipeline consists of three main stages: Initialization, Training, and Evaluation. The initialization stage begins with setting up ROS and SRC, initializing the Gymnasium environment which defines reset, step, and reward functions for each task, and defining learning algorithms. Additionally, it involves configuring the random seed, maximum episode length, network architecture, replay buffer, and other hyper-parameters. The framework can be adapted for imitation learning by adjusting the parameters accordingly.

During the training stage, users first decide on whether to continue or fine-tune other pretrained models, with the model being loaded if applicable. Optional features such as checkpoint auto-save, Tensorboard visualization, and a progress bar may be utilized. Following these preparations, the training process is executed. The test and evaluation stage includes saving/loading the model, predicting actions, controlling the robot, and evaluating performance using specific metrics. This streamlined approach ensures the effective development, training, and evaluation of the agent.

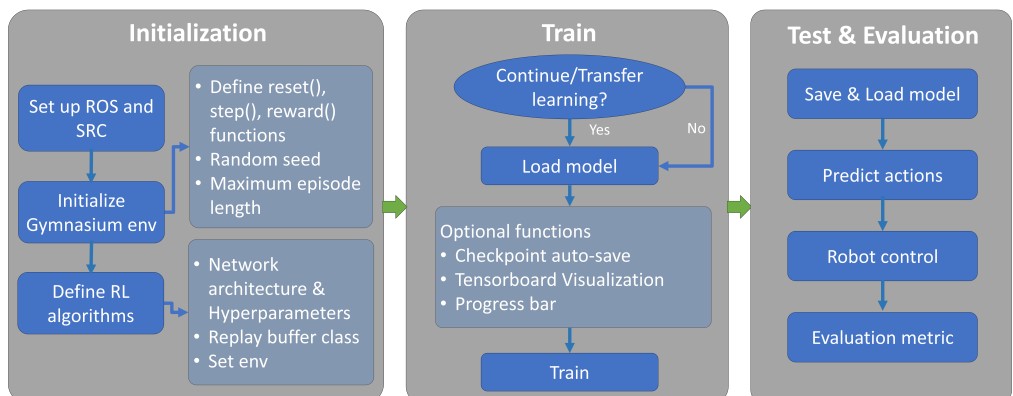

Figure 3: Training pipeline for reinforcement learning.

### 4.2 Teleoperation and Dataset Collection

Given the exploration difficulties inherent in RL implementations, expert demonstrations are often required to mitigate these challenges. SurgicAI provides a comprehensive pipeline for data collection from two primary sources: teleoperation devices and heuristic trajectories generated by Learning from Demonstration (LfD) algorithms [27]. Specifically, the teleoperation data is collected using master tool manipulators (MTMs), while the heuristic trajectories are generated using Dynamic Movement Primitives (DMP) and Locally Weighted Regression (LWR), as described in [27]. For the teleoperation dataset, a human operator uses the teleoperation device to interact with the SRC environment in real time. A recording script captures the PSM joint positions and needle poses in a

rosbag[1]. These recordings enable replaying of trajectories, during which users can collect additional data such as RGB images, depth information, and end-effector positions.

Our environment inherits the benefit of SRC to support various user interfaces for controlling the PSM arms, including graphical user interfaces and devices such as Geomagic and Razer Hydra [18]. Meanwhile, SurgicAI incorporates a standardized data processing pipeline that converts raw data into a format compatible with the Gymnasium API. This process organizes observations, actions, rewards, and terminal states into a structured transition format. This flexibility allows users to switch between different devices for data collection without requiring any modifications, thereby enhancing the efficiency of the data collection process. Additionally, beyond providing a platform for data collection, SurgicAI also offers access to some of the collected trajectories for training purposes, including around 30 human trajectories and 50 heuristic trajectories. These are available on our project website[2] for further use.

### 4.3 Sustained Maintenance Repository

SurgicAI offers a versatile and robust framework for suturing automation, continuously maintained and updated to keep pace with the latest advancements in the field. To create a collaborative environment and facilitate users from different backgrounds, we provide comprehensive instructions for simulator configuration and environment setup, ensuring that users can easily get started. Our documentation includes detailed environment descriptions and demo videos to facilitate understanding and usage. Additionally, we offer a variety of training baselines for different algorithms, supported by extensive RL implementations under the SB3 framework. Our repository also includes various tools and resources to streamline the training. These include processed rollout data from recorded trajectories, customized algorithms, and performance benchmarking scripts. By leveraging these resources, researchers can efficiently develop, test, and validate their algorithms.

## 5 Implementation and Experiment Results

This section covers the implementation of algorithms and experiments, including result analysis. Empirically, we aim to demonstrate the following:

1. The performance of different RL and IL algorithms in suturing tasks, especially the importance of expert demonstration data for successful policy development.

2. The effectiveness of various visual representations in image-guided surgery scenarios.

3. The advantages of our hierarchical task decomposition framework over a single-policy approach.

**Evaluation Metrics:** In this study, we utilized three primary evaluation metrics to assess the performance of low-level policies:

1. Success Rate: The proportion of episodes where the task was successfully completed out of the total number of episodes. This metric reflects the algorithm's ability to perform the task successfully.

2. Trajectory Length: The cumulative distance, measured in millimeters, covered by delta actions during a successful episode. This metric provides insight into the physical efficiency of task completion.

3. Time Steps: The total number of discrete actions taken to complete the task. This metric measures temporal efficiency, indicating how quickly the agent completes the task.

These metrics collectively help evaluate the effectiveness of the low-level policies, where the ideal outcome is a high success rate, minimal trajectory length, and fewer time steps.

**Performance of RL and IL algorithms:** To evaluate the performance of various algorithms, we tested online algorithms like Proximal Policy Optimization (PPO) [43], Deep Deterministic Policy Gradient (DDPG) [44], Soft Actor-Critic (SAC) [45], and Twin Delayed DDPG (TD3) [46], along

---

[1] https://wiki.ros.org/rosbag
[2] https://github.com/surgical-robotics-ai/SurgicAI/tree/main/RL/Expert_traj

Table 1: Performance results of different RL algorithms after 150k training steps, with results averaged over 20 episodes using 5 random seeds. '/' indicates that no successful episodes were collected.

| Algorithm | Reward Type | Criteria | Grasping | Placing | Inserting | Handoff | Pullout |
|---|---|---|---|---|---|---|---|
| PPO | Dense | Success Rate | 0.22 ± 0.10 | / | / | 0.20 ± 0.20 | 0.40 ± 0.49 |
| | | Trajectory Length (mm) | 31.80 ± 3.84 | / | / | 58.54 ± 2.22 | 43.59 ± 1.03 |
| | | Time step | 81.00 ± 8.05 | / | / | 97.09 ± 4.51 | 85.21 ± 4.23 |
| | Sparse | Success Rate | / | / | / | / | / |
| DDPG | Dense | Success Rate | 0.02 ± 0.04 | / | / | / | 0.04 ± 0.04 |
| | | Trajectory Length (mm) | 38.72 ± 2.56 | / | / | / | 40.53 ± 1.56 |
| | | Time step | 87.50 ± 9.50 | / | / | / | 83.94 ± 3.64 |
| | Sparse | Success Rate | / | / | / | / | / |
| SAC | Dense | Success Rate | / | / | / | / | / |
| | Sparse | Success Rate | / | / | / | / | / |
| BC | / | Success Rate | 0.96 ± 0.05 | 1.00 ± 0.00 | 0.97 ± 0.02 | 0.83 ± 0.24 | 0.92 ± 0.09 |
| | | Trajectory Length (mm) | 51.31 ± 14.60 | 66.35 ± 20.36 | 42.43 ± 2.51 | 64.65 ± 3.21 | 49.88 ± 1.74 |
| | | Time step | 100.15 ± 26.02 | 120.87 ± 26.72 | 89.80 ± 3.98 | 113.60 ± 5.15 | 95.80 ± 3.60 |
| TD3 | Sparse | Success Rate | / | / | / | / | / |
| TD3+HER | Sparse | Success Rate | 0.11 ± 0.12 | 0.15 ± 0.09 | 0.05 ± 0.05 | 0.12 ± 0.09 | 0.15 ± 0.14 |
| | | Trajectory Length (mm) | 46.55 ± 7.49 | 55.26 ± 11.25 | 38.71 ± 1.88 | 58.32 ± 10.23 | 45.23 ± 2.46 |
| | | Time step | 86.18 ± 21.85 | 108.62 ± 23.88 | 79.28 ± 3.21 | 95.43 ± 8.32 | 90.46 ± 5.65 |
| TD3+BC | Sparse | Success Rate | 0.85 ± 0.10 | 0.88 ± 0.08 | 0.82 ± 0.15 | 0.79 ± 0.14 | 0.90 ± 0.10 |
| | | Trajectory Length (mm) | 52.45 ± 15.90 | 63.39 ± 15.60 | 46.73 ± 4.59 | 65.98 ± 3.92 | 48.52 ± 1.85 |
| | | Time step | 100.07 ± 18.10 | 116.71 ± 32.88 | 87.00 ± 8.89 | 108.80 ± 4.83 | 94.00 ± 3.90 |
| TD3+HER+BC | Sparse | Success Rate | 0.96 ± 0.06 | 0.97 ± 0.09 | 0.91 ± 0.07 | 0.98 ± 0.02 | 1.00 ± 0.00 |
| | | Trajectory Length (mm) | 47.89 ± 19.25 | 59.27 ± 19.92 | 41.74 ± 2.82 | 61.61 ± 6.15 | 41.88 ± 2.49 |
| | | Time step | 89.61 ± 25.36 | 110.63 ± 31.31 | 83.25 ± 3.30 | 105.00 ± 4.30 | 88.20 ± 4.93 |
| AWAC | Dense | Success Rate | / | / | 0.96 ± 0.06 | 0.98 ± 0.09 | / |
| | | Trajectory Length (mm) | / | / | 44.06 ± 4.23 | 63.40 ± 5.96 | / |
| | | Time step | / | / | 89.32 ± 2.32 | 94.00 ± 10.32 | / |
| BCQ | Dense | Success Rate | 0.94 ± 0.10 | 0.92 ± 0.09 | 0.95 ± 0.08 | 0.96 ± 0.04 | 0.93 ± 0.10 |
| | | Trajectory Length (mm) | 49.22 ± 14.36 | 61.73 ± 9.10 | 42.38 ± 2.31 | 69.78 ± 4.63 | 45.02 ± 3.69 |
| | | Time step | 94.75 ± 19.36 | 117.90 ± 24.46 | 93.00 ± 4.52 | 106.00 ± 7.50 | 93.23 ± 3.21 |
| IQL | Dense | Success Rate | 0.95 ± 0.02 | 0.91 ± 0.09 | 0.94 ± 0.05 | 0.95 ± 0.06 | 0.95 ± 0.08 |
| | | Trajectory Length (mm) | 47.40 ± 15.13 | 67.28 ± 11.52 | 43.46 ± 2.51 | 72.21 ± 3.43 | 44.28 ± 2.34 |
| | | Time step | 92.15 ± 21.57 | 121.85 ± 23.56 | 48.00 ± 4.12 | 110.34 ± 6.21 | 95.21 ± 4.31 |
| CalQL | Dense | Success Rate | / | / | / | / | / |

with ablation tests involving Hindsight Experience Replay (HER) [42] and Behavior Cloning (BC) [47], as well as offline algorithms including Advantage Weighted Actor-Critic (AWAC) [48], Batch-Constrained Q-learning (BCQ) [49], Implicit Q-learning (IQL) [50], and Calibration Q-learning (CalQL) [51]. Both sparse and dense rewards are applied for evaluation. Detailed environment settings and network hyperparameters are provided in Sections 3.2 and A.1, respectively. By default, the BC loss weight during testing is set to 0.5 and we further discuss the impact of the weight in Section A.5.

Table 1 presents the results of each RL algorithm. Pure online RL methods struggled to achieve significant success rates, especially under sparse reward settings. For example, both PPO and DDPG exhibited near-zero success rates for most subtasks, underscoring the challenge of exploration in sparse reward environments. Even with techniques like HER, improvements were limited due to the curse of dimensionality and stringent precision requirements.

Performance improved slightly with dense rewards but often led to suboptimal trajectories, as agents tended to favor the shortest path to task completion. Specific tasks like inserting and handoff posed significant challenges. In the inserting task, agents often adopt strategies with minimal-step completion to minimize reward discounting, occasionally resulting in suboptimal trajectories and missed needle exit points. The handoff task requires precise gripper manipulation to grasp a small needle segment; any collision with the phantom can alter the gripper's pose and lead to failure.

In contrast, integrating BC with RL demonstrated superior performance by closely adhering to expert trajectories. TD3+HER+BC, as demonstrated in A.4, achieved high success rates. While BC alone could attain comparable success rates, the addition of the RL loss optimized the policy, resulting in more efficient trajectories and shorter completion times. For example, in the approaching task, TD3+HER+BC reduced the trajectory length by approximately 5 mm and decreased the completion time by 10 steps compared to BC alone, demonstrating improved efficiency in both spatial and temporal dimensions.

Offline algorithms generally outperformed online ones, particularly in dense reward settings. For instance, BCQ consistently achieved high success rates across all tasks, including 0.95 for the inserting task and 0.96 for handoff. However, algorithms, such as AWAC and CalQL, struggled on tasks like approaching and placing, achieving zero success rates during evaluation. This is because their reliance on accurate advantage or Q-value estimates made them vulnerable in tasks where the offline data was not comprehensive enough to support reliable learning. This performance could be improved with more diverse and abundant offline data.

Table 2: Performance results for each subtask using image-based imitation learning after 30 epochs. Each visual representation was tested under two camera views, with results averaged over 20 episodes using 5 random seeds.

| Task | Criteria | R3M | | CLIP | | ImNet | | RandomNet | |
|---|---|---|---|---|---|---|---|---|---|
| | | front | back | front | back | front | back | front | back |
| Grasp | Success Rate | 0.22±0.14 | 0.35±0.04 | 0.22±0.06 | 0.32±0.06 | 0.08±0.06 | 0.10±0.00 | 0.03±0.05 | 0.03±0.02 |
| | Trajectory length (mm) | 40.81±6.44 | 44.04±7.63 | 50.91±6.03 | 46.32±8.65 | 47.88±8.22 | 49.59±10.04 | 54.85±15.33 | 41.60±4.11 |
| | Time cost (step) | 93.18±19.59 | 94.92±15.75 | 109.63±14.20 | 97.65±22.61 | 89.00±20.66 | 91.83±18.38 | 102.00±31.00 | 74.50±8.50 |
| Place | Success Rate | 0.40±0.07 | 0.22±0.06 | 0.30±0.07 | 0.28±0.09 | 0.27±0.12 | 0.20±0.18 | 0.08±0.06 | 0.02±0.02 |
| | Trajectory length (mm) | 60.80±9.94 | 60.06±8.24 | 70.23±11.43 | 71.89±7.77 | 67.80±9.46 | 77.31±12.60 | 80.70±0.00 | 69.02±2.90 |
| | Time cost (step) | 125.88±17.97 | 134.55±32.86 | 171.60±12.43 | 165.94±29.86 | 181.46±13.32 | 160.46±20.12 | 131.00±0.00 | 120.20±4.83 |
| Handoff | Success Rate | 0.80±0.04 | 0.98±0.02 | 0.62±0.06 | 0.78±0.02 | 0.77±0.06 | 0.83±0.02 | 0.03±0.02 | 0.07±0.03 |
| | Trajectory length (mm) | 56.12±3.72 | 54.09±2.88 | 59.34±4.08 | 58.42±3.90 | 66.70±6.31 | 64.52±3.60 | 71.90±3.61 | 69.02±2.90 |
| | Time cost (step) | 107.98±7.09 | 116.92±5.41 | 125.65±15.16 | 118.47±6.23 | 122.48±17.32 | 117.68±8.78 | 147.50±8.50 | 110.20±4.83 |
| Pullout | Success Rate | 0.82±0.12 | 0.63±0.09 | 0.62±0.02 | 0.60±0.04 | 0.35±0.04 | 0.20±0.07 | 0.10±0.00 | 0.08±0.05 |
| | Trajectory length (mm) | 44.86±2.64 | 50.20±3.40 | 49.42±2.35 | 52.43±2.38 | 50.59±2.87 | 54.10±4.77 | 49.98±1.25 | 46.96±1.39 |
| | Time cost (step) | 76.00 ± 5.58 | 83.47±4.31 | 83.92±3.11 | 94.72±5.17 | 74.95±6.78 | 81.71±12.43 | 78.00±2.58 | 74.25±1.92 |

**Effectiveness of Visual Representations:** In our study of image-based imitation learning, we applied the same evaluation metrics to assess different visual representations, as shown in Table 2. Detailed settings are provided in Section A.2. We evaluated the performance of three popular visual representations: Contrastive Language-Image Pretraining (CLIP) [52], Pretraining Reusable Representations for Robot Manipulation (R3M) [53], and a pretrained ResNet-50 model [54] using the ImageNet dataset (ImNet) [55]. Also, we included a randomly initialized ResNet-50 (RandomNet) as a baseline for comparison. Each model was tested with two camera views: front and back.

A high success rate was consistently observed in the handoff task using the back camera view, suggesting that certain tasks and perspectives provide more favorable conditions for these learning algorithms. R3M slightly outperformed CLIP and ImNet in tasks requiring precise spatial alignment, such as handoff and pullout. Pretrained models consistently outperformed RandomNet in terms of success rate. This is likely because they were trained on large, diverse datasets, enabling them to learn valuable representations such as semantic information and image patterns. These pretrained features can be fine-tuned or directly applied to new domains, significantly reducing computational costs and sample complexity compared to training an encoder from scratch, which can lead to overfitting given the limited scale of our data.

Table 3: Mean success rate (expressed as probabilities) for execution of sequential subtasks. Each result is evaluated over 100 episodes, while each subtask policy from the Hierarchical method is trained with TD3+HER+BC as in Table 1.

| Sequential Tasks | TD3+HER | TD3+HER+BC | Hierarchical |
|---|---|---|---|
| Approaching - Placing | 0.00 | 0.64 | **0.88** |
| Approaching - Inserting | 0.00 | 0.40 | **0.72** |
| Approaching - Handoff | 0.00 | 0.24 | **0.60** |
| Approaching - Pullout | 0.00 | 0.15 | **0.52** |

**Hierarchical Multi-stage Suturing Performances:** While testing the final points, in the configuration of our single-policy approach, we utilized a sparse reward structure, awarding the agent upon the successful completion of each subtask. The detailed result is shown in Table 3. When BC was integrated, expert demonstrations encompassing the entire suturing sequence were employed to guide the agent's actions. Although BC effectively managed individual subtasks, its performance was

limited in longer task sequences. This limitation primarily stems from the accumulation of errors over extended execution, which progressively leads the agent into states that differ substantially from those represented in the demonstration data. Consequently, a distribution shift occurs between the training data and the actual states encountered during the suturing process. In contrast, our hierarchical task decomposition framework demonstrated substantial performance enhancements, achieving over a 50% success rate in the complete suturing procedure. This marked improvement highlights the framework's robustness and efficacy in managing complex surgical tasks.

## 6 Conclusions, Limitations, and Future Plans

In conclusion, SurgicAI provides a comprehensive and standardized pipeline for suturing automation, implementing a robust hierarchical learning framework, and creating a diverse suite of manipulative tasks that benchmark various learning approaches. The results demonstrate that our framework significantly enhances the suturing performance, underscoring its practical impact on surgical automation. The remaining challenges and future work are as follows:

**More realistic simulations:** SurgicAI currently utilizes the simulator version from the 2022-2023 AccelNet Surgical Robotics Challenge [56]. As the challenge evolves, incorporating new models of real phantoms, we will update SurgicAI accordingly. Our datasets and pipelines will be adapted to ensure compatibility with different versions of phantoms and instruments. We are also developing a more realistic rendering pipeline [57] to reduce the domain gap in Sim2Real transfer, as well as injecting the kinematic error [58] of physical robots, ultimately facilitating the fine-tuning and transfer of our policies to real-world suturing.

**More algorithms and tasks:** SurgicAI has been tested with a limited set of algorithms, but still the image-based policies exhibit suboptimal performance, particularly in terms of success rate and stability. The performance also fluctuates significantly with changes in camera perspectives, suggesting that dynamic-view and wrist-mounted cameras could help capture more comprehensive information. To address these issues, we encourage further efforts to improve the robustness of image-based policies. Besides, our current policies rely on Markovian single-step methods, which are inadequate for tasks involving temporal dependencies like pauses or corrections [59]. To overcome this, incorporating advanced algorithms such as the Action Chunking Transformer (ACT) [59] and Diffusion Policy (DP) [60] will be essential. Looking ahead, we aim to integrate vision-language models to improve high-level decision-making, expand our subtask library, and strengthen the system's performance in complex and unexpected scenarios. With more medical simulation environments and applications based on AMBF [61–65], SurgicAI has the potential for deployment across various surgical applications.

**Collaboration:** The surgical robotics challenge, published as an annual competition [56], invites teams from around the world to contribute their trajectories and approaches to suturing automation. To foster a collaborative community where participants can advance algorithms and augment datasets, we aim to further advance SurgicAI via more realistic simulations and more diverse algorithms and tasks. We look forward to collaborating with the broader research community to realize these goals.

## Acknowledgments

This work was supported by NSF AccelNet award OISE-1927354 and an agreement between Johns Hopkins University and Medical Robotics Center (MRC). AM is partially supported by Discovery Award titled Simulation Assisted Navigation for Skull-base Surgery by Johns Hopkins University (JHU). AL is partially supported by the Amazon Research Award, the Discovery Award of the JHU and a seed grant from the JHU Institute of Assured Autonomy. The full team for the development of the underlying SRC simulation and other infrastructures, on which we build our framework, includes Juan Antonio Barragan Noguera, Hisashi Ishida, Haoying Zhou and Dr. Adnan Munawar.

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

# A  Appendix

## A.1  Reinforcement Learning Settings

We perform all our training processes on one Tesla T4 GPU hosted on Microsoft Azure. All sampling during the training are running in headless mode. We implement and modify RL algorithms from the SB3 Library. Both the HLP and LLPs utilize an actor-critic architecture, with the actor and critic networks comprising four fully-connected layers, each containing 256 hidden units and ReLU activations. The LLPs' actions are scaled to the range [-1, 1] using a Tanh activation function in the actor network. Networks are trained using the ADAM optimizer and a default discount factor of 0.99. All LLPs are trained independently in separate Gymnasium environments. The HLP is trained afterward, using the fixed policies of the well-trained LLPs. Besides, When HER is applied for LLP, three additional goals are sampled per episode to enhance the training data, using the 'future' strategy for goal selection. The demonstration data is collected from heuristic trajectories. In our experimental setup, the HLP employs a PPO algorithm. Meanwhile, we test various RL algorithms for the LLPs.

## A.2  Image-Based Imitation Learning Settings

The image-based imitation learning implementation for this hierarchical framework leverages advanced visual representations and proprioceptive data to train both high-level and low-level policies. This approach integrates state-of-the-art techniques in visual perception with imitation learning to enhance task performance.

SOTA visual models such as CLIP, R3M, ImNet are employed to extract key features from high-dimensional image data. These pretrained models are frozen during training to maintain their learned visual processing capabilities. The extracted visual features, combined with proprioceptive data, are processed as the observation input of the downstream policy.

The architecture of the downstream policy for both high-level and low-level policies includes MLPs with three hidden layers, each containing 256 units and preceded by Batch Normalization. The concatenated visual and proprioceptive inputs are processed by the policy network, which generates action outputs scaled to the appropriate range for the corresponding level, similar to the RL implementation.

The training utilizes a dataset that includes visual and proprioceptive data paired with corresponding actions. The high-level policy is updated using cross-entropy loss to generate discrete classifications, while the low-level policy aims to minimize the mean squared error between the expert action and the output action. Both levels use the ADAM optimizer with a learning rate of 0.001.

## A.3  Wall Time Report

We evaluated the average frames per second (FPS) and elapsed time for TD3+HER+BC after reaching 150,000 total time steps in our soft body simulation. The results showed an FPS of approximately 120 Hz, with an elapsed time of around 1,696 seconds. These experiments were conducted in headless mode using a Tesla T4 GPU on the Microsoft Azure platform.

To further accelerate training, we are investigating methods such as parallelizing SRC simulations and sampling data concurrently. While we are still refining these techniques to optimize sampling efficiency, we have provided prototype demonstrations in the repository. This notebook allows users to initiate and manage independent environments in parallel. Details of these developments will be included in future revisions, and we are actively working on integrating SRC with Isaac Sim [66] to enhance overall efficiency.

## A.4  TD3+HER+BC

One SOTA RL technique used for LLPs is TD3 combined with HER, BC, and Q-filter as shown in Figure 4. HER allows the agent to learn from unsuccessful attempts by reinterpreting these attempts as successful ones for different goals, thereby generating additional useful experiences. Specifically, for each real transition, three virtual transitions are created by sampling new goals, significantly increasing the amount of meaningful training data.

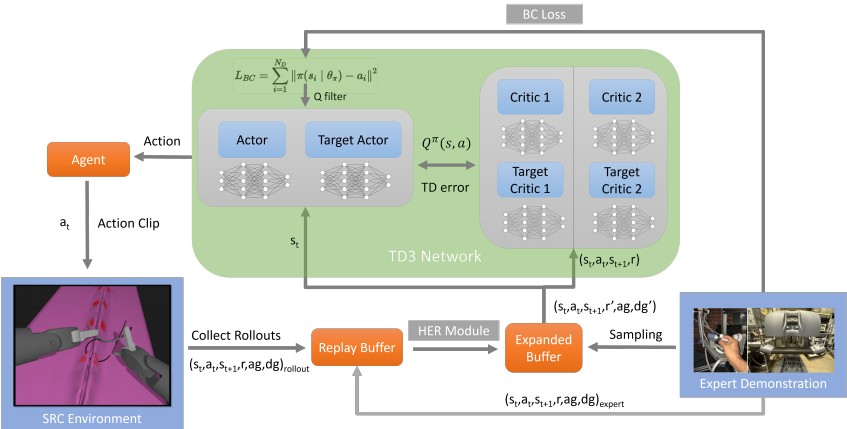

Figure 4: Workflow of the TD3+HER+BC for policy learning. The rollout collected from the SRC environment is stored in the form $(s_t, a_t, s_{t+1}, r, ag, dg)$. Both HER module and expert demonstrations enrich the buffer with additional transitions. The TD3 network, comprising actors and critics, updates its policy based on the rollout from the expanded buffer. The behavior cloning loss $(L_{BC})$ minimizes the error between predicted and expert actions, while using a Q filter to ensure the policy remains close to the expert while avoiding suboptimal solutions.

The training procedure for RL involves several key steps. Initially, both the actor and critic networks are initialized with random parameters. During each environment step, actions are selected with exploration noise, and the resulting state transitions and rewards are observed and stored in a replay buffer. For each gradient step, mini-batches of transitions are sampled from the replay buffer. Target actions are computed with added noise, and the critic networks are updated by minimizing the loss between predicted and target Q-values. Periodically, the actor network is updated using deterministic policy gradients, and the target networks are incrementally updated to track the learned networks.

Expert demonstration data is also integrated into the training process to enhance learning efficiency. These demonstrations are incorporated into the replay buffer, merging with rollout experience collected from the agent's interactions with the simulation environment. The BC loss is calculated using the mean squared error between the actions proposed by the actor network and those from the expert demonstrations. The policy update for TD3 incorporates the BC loss alongside the traditional actor loss, leading to an actor network update informed by both reward-based learning and expert-guided experience.

The Q-filter further enhances the integration of expert demonstrations by selectively incorporating demonstration data into the policy update. It filters out transitions where the critic's value is below a threshold, ensuring that only high-quality transitions influence the policy learning. This improves the overall efficiency and robustness of the training process.

## A.5 Weight Settings of BC Loss

### A.5.1 Calculation of RL Loss

The critic network utilizes Temporal Difference (TD) learning for training, defined by the loss function:

$$L(\theta) = \frac{1}{N_1} \sum_{i=1}^{N_1} \left[ (Q_\theta(s_i, a_i) - y_i)^2 \right]$$

where $y_i$ is the target value for the i-th sample, computed by:

$$y_i = r_i + \gamma \min_{j=1,2} Q_{\theta'_j}(s'_i, \pi_{\phi'}(s'_i))$$

Here, $\gamma$ is the discount factor, $Q_{\theta'_j}$ are the target critic networks, $\pi_{\phi'}$ is the target actor network, and the expectation is approximated by samples from the replay buffer.

The actor network updates through the policy gradient method to maximize expected returns. The actor loss, $J(\pi)$, driven by the policy gradient, is formulated as:

$$\nabla_{\theta_\pi} J(\pi) = \frac{1}{N_1} \sum_{i=1}^{N_1} \nabla_a Q(s, a|\theta_Q)\Big|_{s=s_i, a=\pi(s_i)} \nabla_{\theta_\pi} \pi(s|\theta_\pi)\Big|_{s_i}$$

### A.5.2 Calculation of BC Loss

The BC loss aims to reduce the discrepancy between the agent's actions and the expert's actions, promoting faster initial learning and effective navigation through sparse reward environments. It is mathematically expressed as:

$$L_{BC}(\phi) = \frac{1}{N_2} \sum_{i=1}^{N_2} \|\pi(\phi|s_i) - a_i\|^2 \cdot \mathbf{1}_{Q(s_i, a_i) > Q(s_i, \pi(\phi|s_i))}$$

where each $s_i$ and $a_i$ are states and expert actions from the expert data minibatch, and the indicator function $\mathbf{1}$ ensures that the policy does not simply mimic all expert actions but rather learns to surpass the expert with optimal actions.

### A.5.3 Integration of Losses with Weighted Sum

To effectively combine the RL and BC losses, a weighted combination [31] is used to update the actor policy, where $\lambda_1$ and $\lambda_2$ adjust the influence of the RL and BC losses respectively:

$$\nabla_\phi J' = \lambda_1 \nabla_\phi J(\pi) - \lambda_2 \nabla_\phi L_{BC}$$

The sum of the weights $\lambda_1 + \lambda_2$ is normalized to 1. In our experiment, the variable $\lambda_2$ with values 0.2, 0.5, and 0.8, is employed to examine its dependency on expert demonstrations and its effects on the learning dynamics as shown in Table 4. Additionally, this variation aims to strike a balance between exploration driven by environmental interactions and the exploitation of established, successful behaviors.

Table 4: Mean success rate (expressed as probabilities) for each subtask using different BC loss weights and expert trajectory numbers after 150k training steps. Each result is evaluated over 100 episodes.

| Task Name | BC Loss Weight | Trajectory Numbers | | |
|---|---|---|---|---|
| | | 10 epi. | 20 epi. | 30 epi. |
| Approaching | 0.2 | 0.53 | 0.65 | 0.79 |
| | 0.5 | 0.61 | 0.76 | 0.90 |
| | 0.8 | 0.60 | 0.87 | 0.94 |
| Placing | 0.2 | 0.41 | 0.70 | 0.84 |
| | 0.5 | 0.50 | 0.81 | 0.90 |
| | 0.8 | 0.55 | 0.92 | 1.00 |
| Inserting | 0.2 | 0.54 | 0.66 | 0.80 |
| | 0.5 | 0.62 | 0.77 | 0.88 |
| | 0.8 | 0.61 | 0.88 | 0.95 |

### A.6 Heuristic Trajectory collection

### A.6.1 Data Collection and Preprocessing

**Data Collection** To obtain expert data, we collect the demonstrations from human subjects. We ask the human subjects to perform a single-throw suture procedure in the simulation environment [18] using a dVRK MTM. We record all the kinematic and dynamic data during the teleoperation. Then, taking advantage of the great repeatability of AMBF [41], we can steadily replay the movements and select specific kinds of data we would like to collect. The whole data collection framework follows the pipeline developed in [27] as shown in Figure 5.

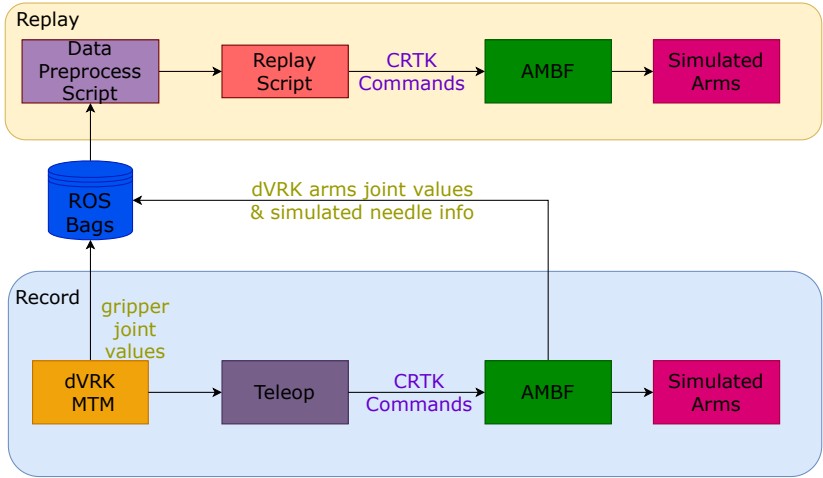

Figure 5: Expert demonstration data collection pipeline.

## Data Decoupling

The whole suturing procedure can be divided into five subtasks as stated in the manuscript. Therefore, we need to decouple the raw recorded data corresponding to the five subtasks. We design a Python script to manually dissect the raw recorded data when replaying.

**Data Filter** To obtain solid motion data, it is also important to exclude the static states to reduce the variance. Therefore, after data decoupling, we also implement a velocity-based filter to the segmented sub-dataset. We set the thresholds of 0.2 mm/s and 0.1 deg/s for linear and angular velocity lower limits. For any data points which have velocities smaller than the lower limits, the data points will be considered to be static and removed from the dataset.

### A.6.2 Imitation-learning based solution

We use the learning from demonstration algorithm in Zhou et. al. [27] as our baseline. In Zhou et. al., their algorithms mainly focus on the automation of the needle insertion and extraction subtasks. In their work, they use dynamic movement primitives (DMP) [67–70] to achieve the learning from demonstration (LfD) algorithm.

The whole pipeline achieving the suturing task automation in Zhou et. al. can be summarized in the following algorithm:

---
**Algorithm 1:** Suturing Automation using Learning from Demonstration

---
1. Data Collection, data decoupling and data preprocessing.
2. Select one needle trajectory from expert demonstrations, calculate the optimal learning weights for DMP using locally weighted regression[71].
3. Select the desired start and goal states for new trajectory generation.
4. Generate the needle trajectory using the LfD algorithm [27] given the learning weights, start and goal states.
5. Convert the needle trajectory into the PSM trajectory using the inverse kinematics.
6. Send the PSM trajectory data point to AMBF simulator for simulated arm movements.

---

