# OpenReview forum: "SurgicAI: A Hierarchical Platform for Fine-Grained Surgical Policy Learning and Benchmarking"
_NeurIPS.cc/2024/Datasets_and_Benchmarks_Track — NeurIPS 2024 Track Datasets and Benchmarks Poster_

### Official Review · Reviewer_D6Tc · 2024-07-19
**Great work, clear paper, repository needs some work**

**Rating:** 6
**Confidence:** 4
**Clarity:** Yes

**Review:**

Overall, the authors did substantial work and the paper is clearly written. However, this work could greatly benefit from better organization of the repository.

Pros:
* Diverse set of suturing sub-tasks and novel hierarchical task decomposition were implemented using familiar RL libraries (Gymnasium + Stable Baselines3)
* The environment, tasks, and experiments are clearly described
* The expert trajectory data are provided for the research community

Cons:
* Repository lacks clear organization and comprehensive documentation
* Difficulty in mapping paper results to specific code implementations

**Strengths:**

* Focus on complex, multi-stage surgical tasks like suturing, which presents significant challenges in robotics automation.
* Implementation of a hierarchical task decomposition framework was very interesting, which can be later used with human-ai collaboration and/or VLM-based planning
* Clear paper presentation with well-structured experiments comparing various reinforcement and imitation learning approaches.
* Provision of expert trajectory data, which could be valuable for other researchers in the field.

**Additional Feedback:**

* Line 146: 1 otherwise → “-1” otherwise?
* Is the directory in the instruction correct? https://github.com/surgical-robotics-ai/SurgicAI/blob/main/surgical_robotics_challenge/docs/teleoperation_supplemental.md#run-teleoperation

**Correctness:**

The authors stated that the primary contributions are: (1) standardized pipeline, (2) hierarchical task decomposition, (3) diverse suite of tasks, and (4) benchmark of performance.

The SurgicAI repository contains the pipeline for collecting and preprocessing expert trajectories, which seems to be similar to the code in the https://github.com/surgical-robotics-ai/surgical_robotics_challenge. However, the authors also stated that this pipeline was developed in Zhou et al., 2024. It would be helpful to clarify how these pipelines differ or if they are the same.

The other claims seem to be correct. However, it would be more helpful if the implementations for the hierarchical task decompositions and evaluation metrics were clearly documented in the repository.

**Documentation:**

The authors have been running several robotic surgical challenges, workshops, and tutorials about this platform, so documentation is great overall. The reproducibility of the current repository could greatly benefit from improved organization and instructions.

**Limitations:**

The authors have been running several robotics surgical challenges, so the limitations are well described.

**Opportunities For Improvement:**

* There is a lot of stuff in the repository. It would hugely benefit from improved organization and correct documentation. Also, a Dockerfile that works right away with a tutorial for training, evaluation, and demo generation will substantially boost reproducibility.
* Wall time for training policies is not reported, which is important for assessing computational requirements.
* Training could potentially benefit from domain randomization or curriculum learning.

Questions about the repo:
* Are the best-performing models (e.g., TD3+HER+BC) available in the repository? From the repo, I could only find TD3_BC models under RL/Evaluation_model.
* How can the demo video be reproduced? Is this from the hierarchical method? It would be great if it can be reproduced.
* How do the three evaluation tasks in the code (RL/evaluate.py) correspond to the results presented in the paper?

**Relation To Prior Work:**

Yes

**Summary And Contributions:**

SurgicAI proposes a comprehensive framework for data collection and benchmarking in surgical policy learning on suturing, a complex and challenging surgical process. Key contributions include:
* A set of (sub-)tasks/environments for suturing policy training, along with scripts for RL/IL training and evaluation
* Presentation of hierarchical task decomposition for multi-stage surgical procedures
* A set of scripts for expert trajectory collection and utilization

---

> ### Author Rebuttal · Authors · 2024-08-17
>
> Thank you very much for your valuable feedback! Below are our responses to your comments:
>
> 1. **Repository Organization and Documentation:** We have updated our GitHub repository with more detailed documentation and tutorials. It is now more user-friendly, making it easier to reproduce our models. Additionally, we have provided [pretrained models](https://github.com/surgical-robotics-ai/SurgicAI/tree/main/RL/Evaluation_model) that can be used to regenerate the demo videos.
>
> 2. **Wall Time Reporting:** Please refer to the general response for this part.
>
> 3. **Domain Randomization and Curriculum Learning:** We greatly appreciate your suggestions regarding domain randomization and curriculum learning. We agree that domain randomization is crucial for our future sim-to-real efforts. We have made preliminary attempts to incorporate these concepts into our environment setup. For instance, the `camera_view_reset` function introduces noise to randomize the camera view, and the `light_random` function varies the light source positions to simulate the changing shadows and lighting conditions encountered in real surgical scenarios. Additionally, we have transferred and tested our training pipeline on new phantoms, as shown in the demo videos in our [repository](https://github.com/surgical-robotics-ai/SurgicAI). Our future work will focus on adding more domain randomization factors, such as changing backgrounds and dynamic phantom positions. Regarding curriculum learning (CL), we have uploaded a [CL wrapper](https://github.com/surgical-robotics-ai/SurgicAI/blob/main/RL/CL_env.py) for our environment that gradually increases task difficulty and reduces error tolerance in the subtasks. There is a [guidance script](https://github.com/surgical-robotics-ai/SurgicAI/blob/main/RL/PPO_train_curriculum.ipynb) as well on how to use this wrapper. This should help mitigate exploration difficulties for RL, and we plan to evaluate its performance in future work.
>
> ### Responses to Your Questions:
>
> 1. **Algorithm Configuration and TD3+HER+BC:** We have uploaded algorithm configuration files in the repository, allowing you to set the parameters for each algorithm. For TD3+HER+BC specifically, in the SB3 library, the HER module is implemented as a replay buffer class rather than as an algorithm. You can find clearer definitions in the [configuration files](https://github.com/surgical-robotics-ai/SurgicAI/blob/main/RL/algorithm_configs_online.py).
>
> 2. **Reproducing the Demo Video:** To reproduce the demo video, you first need to complete the training for all five subtasks and then train the high-level policy (we also provide you some pretrained models as well). The evaluation process of the high-level policy will invoke these well-trained low-level policies. We have uploaded additional [instructions](https://github.com/surgical-robotics-ai/SurgicAI/blob/main/README.md) to the GitHub repository to assist with this process.
>
> 3. **Clarification on Evaluation Scripts:** The script you mentioned appears to be from the SRC repository and was not used to evaluate our policy performance. We have now uploaded a comprehensive [model-evaluation script](https://github.com/surgical-robotics-ai/SurgicAI/blob/main/RL/Model_evaluation.py) that provides an interface to evaluate any algorithm or task. This should give you more insight into how our policies are evaluated.
>
> 4. **Comment on Your Feedback:** Regarding your feedback, thank you for pointing out the typo. Indeed, on line 146, the agent should receive a reward of -1. We will modify the typo in our final paper. Additionally, we've included more detailed [instructions](https://github.com/surgical-robotics-ai/SurgicAI/tree/main/Teleop_collection) on using the teleoperation device (specifically the MTM) for data collection.

---

### Official Review · Reviewer_SDfj · 2024-07-22
**Review of "SurgicAI: A Fine-grained Platform for Data Collection and Benchmarking in Surgical Policy Learning"**

**Rating:** 6
**Confidence:** 3

**Review:**

Please see Strengths and Opportunities for Improvement below.

**Strengths:**

- The paper introduces a standardized environment for each stage of a complex surgical task, suturing. Surgical robotics tasks are a promising application of robot learning, and the development of additional standardized environments and benchmarks can assist progress in this area.
- The task definitions are benchmarked using a number of sensible baselines for reinforcement learning and imitation learning.
- The work introduces a method for hierarchical decomposition of the suturing task into subtasks, which when combined with a hierarchical policy, is able to significantly improve the performance on longer horizon tasks.
- Ablation experiments demonstrating the effects of using hindsight experience replay and auxiliary behavior cloning losses help the reader understand challenges and effective solutions for these environments.

**Additional Feedback:**

A couple of minor points of feedback:


- For the title of the paper, I might suggest removing or rephrasing the term “fine-grained”. It’s not immediately clear to me what is meant by “fine-grained platform”. The first thing that comes to mind is that the task themselves require fine-grained manipulation, but in this case I might suggest using it to describe the tasks or policy learning e.g. “Fine-Grained Surgical Policy Learning”. My other guess is that it refers to breaking down the long horizon task into multiple subtasks. In this case I might suggest rephrasing to an adjective which more directly gets at this point.


- As a nitpicky and completely optional suggestion, I might recommend rephrasing the four paragraphs in Section 1 after “The primary contributions of SurgicAI are as follows”:. This is entirely due to personal preference as those paragraphs sound similar to that written or edited by a language model, which is slightly distracting. If this is indeed the case, I understand that the NeurIPS submission policy allows for this but I would suggest paraphrasing a bit here. If the text was written entirely by a human, I sincerely apologize.

**Clarity:**

The paper is in general written quite clearly and is easy to understand. There are a few minor typos that do not interfere with the reader’s understanding of the work e.g. L162: “with with positional”. I recommend the authors to carefully proofread the manuscript.

**Correctness:**

Generally the experimental setup seems reasonable, however the RL experiments are only run with one random seed each. The authors explain that there are many evaluation episodes used for each success rate statistic, but for RL the variance of policy performance can be quite large between different random seeds. I would suggest using a minimum of 3 random seeds and optimally 5.

**Documentation:**

I believe there is sufficient detail to support reproducibility. The linked GitHub repository appears to provide sufficient documentation and installation instructions, although I did not test it myself.

**Ethics:**

I don’t suspect there are any ethical concerns with the submission.

**Limitations:**

The authors have reasonably addressed the limitations of their work in the last section of the manuscript. I encourage them also to discuss any computational aspects (e.g. simulation speed) of the limitations if any. I agree with the authors in that I do not foresee significant negative societal impact of this work.

**Opportunities For Improvement:**

- The paper discusses several prior works that use GPU-accelerated simulators, but it does not describe the computational cost of running the simulation pipeline that it describes. While this is likely a property inherited from the SRC simulation, I still think it would be helpful to describe this in the text (e.g. average number of simulation steps/second) and compare it to the other works described in the related works section, such as Surgical Gym, Surrol, and ORBIT-Surgical. It would also be helpful to report in wall clock time how long it takes to train the RL policies, as well as the amount of time it takes for collecting human demonstrations for imitation learning.
- Unlike prior works, this work does not validate the potential effectiveness of this platform in transferring learned policies to the real world. While I don’t think this is absolutely necessary as this is not claimed as one of the main contributions of the paper, it would significantly improve the value of the work if learned policies could be transferred to the real world, or it could somehow be determined that success of a certain method on this benchmark could correlated with performance using real data on a real system.
- The work does not appear to make innovations on the simulation side itself, with most simulation properties directly inherited from the SRC simulation. The suturing task itself also seems to have been introduced in prior work. Thus, one opportunity for improvement is to introduce additional simulation tasks beyond suturing, particularly those that take advantage of soft-body simulation.
- It would be interesting for the authors to discuss in greater detail what they believe the main research questions that the platform, as it is, should be used now to explore. The authors discussed the performance of existing methods on the simulation platform, but what are the critical problems that the community as a whole should investigate now? This is especially important given that the performance of behavior cloning on each of the subtasks seems to be nearly 100%.
- While it appears that both sparse and dense rewards are defined for each of the subtasks, RL methods are never benchmarked using the dense rewards, only the sparse rewards. This would be helpful for completeness.

**Relation To Prior Work:**

I think there are a few specific details in regards to relation to prior work which could be made more clear.

Firstly, in L73: “Additionally, most of these platforms lack soft body simulation, which limits their physical realism.” I think it would be better to directly point out which of the platforms specifically have or do not have soft body simulation, and that the described platform does have soft body simulation. This will help to clarify how the work differs from dVRL, SurRoL, and Surgical Gym.

The discussion with ORBIT-Surgical beginning in L75 also is slightly misleading. The authors argue that “In contrast, SurgicAI offers a comprehensive solution by addressing these limitations and providing robust support for complex suturing procedures.” However, it appears that the addition of complex suturing procedures is inherited from the use of the simulator from the SRC, which is not a new contribution in this paper.

**Summary And Contributions:**

This paper introduces SurgicAI, which is a simulation environment for learning to perform surgical tasks using imitation learning and reinforcement learning. SurgicAI is built upon the AccelNet Surgical Robotics Challenge (SRC), and introduces several contributions. It integrates imitation learning from demonstrations and reinforcement learning with the simulation into a complete pipeline, provides decomposition of the surgical tasks into subskills, and benchmarks performance of popular IL and RL methods.

---

> ### Author Rebuttal · Authors · 2024-08-17
>
> Thank you for your detailed feedback and for highlighting key areas for improvement. We appreciate your insights and would like to address your comments as follows:
>
> 1. **Computational Cost and Simulation Performance:** As mentioned in our general response, we conducted a wall time test, recording the average FPS and elapsed time for TD3+HER+BC at 150k total time steps. While our current simulation speed is slower than GPU-accelerated simulators like Isaac Sim, we are actively working on integrating SRC with Isaac Sim to enhance performance. We have also made progress in parallelizing simulations to accelerate data collection, as detailed in our repository.
>
> 2. **Sim-to-Real Transfer and Validation:** We recognize the importance of sim-to-real transfer in validating our platform. We are currently focusing on transferring learned policies to our new SRC environment, which features more realistic medical tools and phantoms. Demo videos of this work are available in our [GitHub repository](https://github.com/surgical-robotics-ai/SurgicAI). We are committed to further refining these methods and providing comprehensive sim-to-real validation in future studies.
>
> 3. **Contributions Beyond SRC and Task Expansion:** While SurgicAI does not introduce new simulation assets, our work emphasizes advancing high-level decision-making and path planning for autonomous suturing. We have detailed these contributions in the general response and are open to deploying our system in new scenarios from the SRC community.
>
> 4. **Main Research Challenges:** As discussed in the general response.
>
> 5. **Experimental Completeness:** Thank you for your advice on enhancing the completeness of our experiments. We have expanded our algorithm testing to include both dense and sparse rewards for select algorithms, along with a combination of online and offline RL methods. To ensure reproducibility and robustness, we conducted RL training with five random seeds. Detailed results are available in the attached file. We will add this table to our paper, which should show a more comprehensive evaluation of our methods.
>
> 6. **Relation to Prior Work:** Regarding your comment on L73, we would like to point out that simulations such as dVRL and Surgical Gym do not provide soft body simulation capabilities. However, recent work has demonstrated soft body development in SurRoL [1]. As for the comment on L75, as discussed in our general response, our contribution lies in bridging the machine learning algorithm development with SRC and applying the hierarchical framework as a solution for long-term complex tasks. We will address the misunderstandings and correct the typos you mentioned in our final paper.
>
> 7. **Title:** Considering your suggestions on the title, we propose our title as “SurgicAI: A Hierarchical Platform for Fine-Grained Surgical Policy Learning and Benchmarking.” We believe this title more accurately reflects the key features of our work.
>
> 8. **Writing of the Contribution List:** The sentences you mentioned were not generated by the language model, but we will make them more natural in our final paper.
>
> [1] Yang, Zhenya, et al. "Efficient Physically-based Simulation of Soft Bodies in Embodied Environment for Surgical Robot." arXiv preprint arXiv:2402.01181 (2024).

---

> > ### Comment · Reviewer_SDfj · 2024-08-18
> > **Response to author rebuttal**
> >
> > Dear authors,
> >
> > Thank you for your detailed responses to my questions and requests for clarification. I very much appreciate the additional information and experiments, particularly about the computational cost, experimental completeness, relation to prior work, writing of the contribution list, and title, which have resolved my concerns about those points.
> >
> > I have a few follow-up clarifying questions which I was hoping you would be able to answer about some of the other points:
> >
> > **Re: Contributions Beyond SRC and Task Expansion:** Could you please specify which specific detailed components of the simulation environment are contributed in this work? For instance, is the low-level action space used in SurgicAI different from that used in SRC? Were the environment reward definitions newly introduced in this work? What about the simulation setup for each step of the suturing task (maybe not for this one if we aren't introducing new simulation assets here)?
> >
> > **Re: Main Research Challenges:** Thank you for clarifying this point. It indeed seems that learning image-based policies is still a challenge. Would it be possible to evaluate the performance also of using a randomly initialized visual encoder and training the encoder together with the policy for this setting rather than using a pretrained and frozen encoder? It may be that those models (R3M, CLIP, ImgNet) were trained on a very different data distribution from what is seen in this environment, and since they are frozen during imitation learning, the model is unable to ever extract useful features for this task. Apologies for not asking about this in my original review.
> >
> > Thank you again for your responses and clarifications!

---

> > > ### Author Response · Authors · 2024-08-21
> > >
> > > Thank you for your response. To further answer your questions:
> > >
> > > **Contributions Beyond SRC and Task Expansion**
> > >
> > > In SurgicAI, we defined our actions as delta-controlled, meaning that at each timestep, the agent moves forward by an incremental step. The agent learns which action to take at each step based on this control logic, which is newly implemented in SurgicAI. This approach allows for finer granularity in control, enabling the agent to make more precise adjustments and better respond to dynamic environments. In contrast, in SRC and other well-established robotic systems, the community typically provides APIs like `servo_jp` or `servo_cp` that move either the joints or the end-effector to desired absolute positions.
> > >
> > > Additionally, since our work is the first to integrate Gym-env into SRC, all components related to robot learning—such as reward functions, state reset mechanisms, action steps, termination conditions, etc.—are newly introduced in our framework. For further details, please refer to our [environment setup script](https://github.com/surgical-robotics-ai/SurgicAI/blob/main/RL/subtask_env.py). Moreover, we also introduce a hierarchical solution for long-horizon tasks, where the definitions of each subtask and the transition states are new contributions within our work. To clearly distinguish what new assets are brought into SurgicAI, SRC is set as a submodule in our [updated repository](https://github.com/surgical-robotics-ai/SurgicAI).
> > >
> > > **Main Research Challenges**
> > >
> > > Thanks for your suggestion. We fully agree that a randomly initialized visual encoder is a valuable baseline when evaluating the effectiveness of other encoders. To this end, we conducted experiments with a non-pretrained ResNet50, training it alongside the downstream policies. The results, as shown in the [attached link](https://livejohnshopkins-my.sharepoint.com/:b:/g/personal/jwu220_jh_edu/EWCn1gk4s2RLjxl2T59lFJUBN3agC2ikRrHCZCsxYrabUw?e=hdas9K), indicate that this network, when trained from scratch, does not outperform other pre-trained networks. We will add this baseline to our revision.
> > >
> > > We also acknowledge that the visual encoders we used were pre-trained on datasets with different distributions compared to our specific scene. However, this does not necessarily imply they are ineffective. In fact, pre-trained models, especially those trained on large and diverse datasets, have already learned valuable representations, such as semantic information and image patterns, which can be fine-tuned or directly applied to new domains. In our work, R3M was specifically designed for robotic manipulation, making it a natural fit for our scene. Additionally, studies like [1] and [2] have demonstrated the effectiveness of CLIP in robot policy, while earlier work like [3] and [4] has confirmed the success of pre-trained ImageNet encoders in robot learning. All of these applications keep these encoders frozen during the training. By leveraging these general pre-trained encoders, we can significantly reduce the computational cost and sample complexity associated with transferring or fine-tuning policies to new scenarios.
> > >
> > > **References**
> > >
> > > 1. Shridhar, Mohit, Lucas Manuelli, and Dieter Fox. "Cliport: What and where pathways for robotic manipulation." *Conference on robot learning.* PMLR, 2022.
> > >
> > > 2. Khandelwal, Apoorv, et al. "Simple but effective: Clip embeddings for embodied ai." *Proceedings of the IEEE/CVF Conference on Computer Vision and Pattern Recognition.* 2022.
> > >
> > > 3. Radosavovic, Ilija, et al. "Real-world robot learning with masked visual pre-training." *Conference on Robot Learning.* PMLR, 2023.
> > >
> > > 4. Shah, Rutav, and Vikash Kumar. "Rrl: Resnet as representation for reinforcement learning." *arXiv preprint arXiv:2107.03380 (2021).*

---

> > > > ### Comment · Reviewer_SDfj · 2024-08-21
> > > > **Response to authors**
> > > >
> > > > Thank you for the response and additional experiments. These largely elucidate the questions that I had about contributions beyond SRC and the performance of a visual encoder trained from scratch, although I'm quite surprised that a visual encoder trained together with the policy performs so poorly compared to the pretrained encoders. Can you please provide some additional details about the imitation learning experiment? For instance, how many demonstrations are used for training the policy for each task? How many environment transitions does this equate to? These details are very important for understanding the results, but I'm not able to find them in the manuscript or rebuttal pdf.
> > > >
> > > > Assuming the authors can provide more details about the imitation learning experiment, I'm increasing my score to 6, although I am hesitant to give a higher rating given the scope of current contributions on the simulation benchmark side compared to the SRC simulator.

---

> > > > > ### Author Response · Authors · 2024-08-21
> > > > >
> > > > > Thank you for your feedback. The underperformance of the visual encoder trained from scratch is primarily due to the policy overfitting to our dataset, which may lack sufficient diversity for generalization. Below is a table that summarizes the training and testing loss after 40 epochs of training on our updated dataset, specifically for the front view in the Approaching task. We would like to point out that achieving generalization with a model trained from scratch may require additional careful hyperparameter tuning and extensive dataset, which could be resource-intensive. Pre-trained models, on the other hand, offer more generalizable representations that inherently mitigate the risk of overfitting. We will incorporate this discussion into the paper.
> > > > >
> > > > > | Model   | Training Loss | Testing Loss |
> > > > > |---------|---------------|--------------|
> > > > > | R3M     | 0.4750        | 0.5263       |
> > > > > | CLIP    | 0.4612        | 0.5389       |
> > > > > | ImgNet  | 0.3266        | 0.6164       |
> > > > > | Random  | 0.1041        | 0.6463       |
> > > > >
> > > > > For the imitation learning experiment, each subtask in our image-based dataset includes 50 demonstrations, leading to the following transitions:
> > > > > - **Approach:** 4,696
> > > > > - **Place:** 6,792
> > > > > - **Handoff:** 5,791
> > > > > - **Pullout:** 4,156
> > > > >
> > > > > For more details about our dataset, please refer to the [dataset link](https://github.com/surgical-robotics-ai/SurgicAI/tree/main/Image_IL/SRC_img_data) in our repository. We hope this clarifies the details you requested.

---

> > > > > > ### Comment · Reviewer_SDfj · 2024-08-26
> > > > > > **Thank you**
> > > > > >
> > > > > > Thank you for the details and summarizing the imitation learning results, this does clarify the details I requested.

---

### Official Review · Reviewer_LYxJ · 2024-07-27
**Review for SurgicAI**

**Rating:** 7
**Confidence:** 3
**Correctness:** Yes, the claims in this work are corr…

**Review:**

The authors provide a framework to study the performance of imitation learning and reinforcement learning on a set of surgical tasks. The paper is well-written and has high clarity with extensive evaluation for both lower-level subtasks and hierarchical planning. The authors also provide architectural ablations, such as considering alternative forms of visual features, showing their transfer to this domain (non-neglible difference from a random initialization). Additionally, the authors provide a reproducible data collection setting, allowing for others to collect demos using their teleop system as well as providing appropriate data filtering/decoupling.

**Strengths:**

- Provides compatibility with open source frameworks such as Gym and stable baselines.
- Provides a clear framework for dataset collection and filtering/decoupling

**Additional Feedback:**

Good paper overall. Excited to hear about the questions I had in the opportunities for improvement section.

**Clarity:**

Yes, the paper is well written with clear figures and a description of the contributions of the authors.

**Documentation:**

Yes, the authors provide a link to a github repository with these details.

**Ethics:**

No ethical concerns present.

**Limitations:**

Yes, limitations are adequately discussed in this work.

**Opportunities For Improvement:**

- The dataset seems to be saturated by imitiation learning and td3bc + her + bc with the hierarchical system, allowing for limited potential improvement in this benchmark. Potentially provide tasks/domains where improvement can be done and discuss what challenges these approaches face in this setting
- Approaches in offline RL might provide for a better initialization for online RL, given the promise of imitation learning. I understand that TD3 + BC was considered in this work but may want to consider alternative algorithms such as RLPD, Cal-QL, and IQL, which are more sample efficient.
- Given the limited demos (30) as seen in Table 4, concerned slightly about the robustness of the system. Could you provide more details about the evaluation approach and how it compares to the demos that were collected?

**Relation To Prior Work:**

Partially, the environment is based on the AccelNet Surgical Robotics Challenge. It would bring more clarity to the paper if the authors provide detail on what changes were needed from the robotics challenge to construct this benchmark.

**Summary And Contributions:**

The authors provide a Surgical Benchmark for policy learning.  They emphasize that alternative approaches are limiting as they aren't photorealistic and/or perform complex long-horizon surgical tasks. The authors state that the data for this project is collected with a teleop system, utilizing an environment based on the AccelNet Surgical Robotics Challenge which has simulated PSMs from the da Vinci Surgical System.  The authors provide compatibility with Gym and Stable Baselines to effectively test different approaches using RL or imitation learning. The authors also provide a hierarchical framework to select a limited set of low-level tasks: Grasping, Placing, Inserting, Handoff, and Pullout and show benefits vs learning fully end-to-end citing reasons of failure being the sparse rewards over a long horizon. For empirical validation, the authors test imitation learning and reinforcement learning policies and consider different pre-trained visual representations as an initialization.

---

> ### Author Rebuttal · Authors · 2024-08-17
>
> Thank you for your insightful review! We have addressed your concerns as follows:
>
> 1. **Saturation of the Benchmark and Potential for Improvement:** As discussed in our general response, our state-based policy leverages ground-truth data with relatively low-level information, making it easier to learn compared to image-based policies. There is still significant room for improvement, particularly in the context of image-based policies. Our current work primarily serves as a verification of our system’s feasibility, and we acknowledge that further efforts are required to enhance long-horizon performance and conduct sim-to-real testing.
>
> 2. **Application of Offline RL:** We greatly appreciate your suggestion to apply offline RL methods. Pretraining the network with offline demonstration data indeed facilitates training efficiency. To incorporate this, we have extended our supported open-source library to `d3rlpy` and have implemented several offline RL algorithms, including CalQL, AWAC, BCQ, and IQL. Preliminary results are available in Table 1 of the attached file.
>
> 3. **Evaluation Details:** Regarding the evaluation methodology, we have enhanced our benchmark indices to include success rate, trajectory length, and time step as demonstrated in our general response. To test the robustness and reproducibility of RL algorithms, we trained the models using five random seeds. Each episode begins with state randomization, and the success criteria are detailed in the general response.

---

### Author Rebuttal · Authors · 2024-08-17

We would like to thank all reviewers for their time and constructive feedback on our paper. Since some reviewers referred to similar concerns, we would like to make a general response to address these questions.

1. **Clarification of Contributions Beyond SRC Simulation:** We acknowledge that our previous work is built upon SRC, but we want to clarify our contributions beyond SRC here. The SRC provides an excellent low-level infrastructure for research, offering realistic rendering, precise modeling of medical instruments, and an API for basic raw data querying and robot control, along with support for various teleoperation devices. Our work, however, focuses on high-level motion planning and robot learning. This includes constructing a Gym-compatible environment with low-level APIs, establishing a pipeline for collecting trajectories (from humans or scripts with synchronized data from sensors), and processing raw data into trainable datasets compatible with our RL and IL algorithms. Additionally, we employ a comprehensive pipeline to train and benchmark the performance of various algorithms (both state-based and image-based). We will make this more clear in our revision.

2. **Wall Time for Training:** We recorded the average frames per second (FPS) and elapsed time for TD3+HER+BC when reaching 150k total time steps. With our soft body simulation, the FPS is approximately 120 Hz, and the elapsed time is around 1696 seconds. These tests were conducted with headless mode on a Tesla T4 GPU hosted on Microsoft Azure. We will add our hardware details and training time information in our revision. To further improve training speed, we are exploring methods such as running parallel SRC simulations and sampling data simultaneously. While we are still developing these features to enhance sampling efficiency, we have provided prototype demos in the `multi_env_test.ipynb` in the repository. This allows users to launch and control independent environments separately. We will provide these details in our revision, and we are actively working on integrating SRC to Isaac to further boost efficiency.

3. **More Comprehensive Experimental Results:** Following reviewers’ suggestions, we conducted additional experiments to benchmark the methods as shown in Tables 1 and 2 of attached file. We now use 5 random seeds to generate error bars for the results, include both state-space and image-based RL algorithms, and include more comprehensive evaluation metrics that matter in practice. In particular, our evaluation criteria involves: (1) success rate, defined as the number of successful episodes divided by the total episodes; (2) trajectory length, measured as the cumulative transition driven by delta actions during the successful episode; and (3) time step, defined as the number of steps to complete the task.

4. **Remaining Research Challenges:** As shown in Tables 1 and 2 in the attached file, the performance of image-based policies remains suboptimal. Specifically, the performance is unsatisfactory in terms of success rate and stability. We can see that the success rate for image-based methods is generally low with considerable variance across different tasks, particularly in challenging tasks like “Approaching”. Moreover, the effectiveness of the policy is highly sensitive to the camera view; varying camera perspectives can lead to inconsistent performance on the same task. A potential solution is to introduce multi-view cameras and wrist-mounted cameras to capture more comprehensive information. Therefore, we encourage the community to further improve the effectiveness and robustness of image-based policy. On the other hand, the policies we applied primarily rely on Markovian single-step methods, which may struggle with tasks requiring an understanding of temporal dependencies, such as pausing or making corrections [1]. We recognize this limitation, and this problem can be mitigated by incorporating state-of-the-art algorithms like the Action Chunking Transformer (ACT) [1] and Diffusion Policy (DP) [2]. Exploring these approaches will be our future work.

### References

- **[1]** Zhao, Tony Z., et al. "Learning fine-grained bimanual manipulation with low-cost hardware." arXiv preprint arXiv:2304.13705 (2023).
- **[2]** Chi, Cheng, et al. "Diffusion policy: Visuomotor policy learning via action diffusion." arXiv preprint arXiv:2303.04137 (2023).

---

### Decision · Program_Chairs · 2024-09-26

**Decision:**

Accept (Poster)

**Comment:**

The paper presents SurgicAI, a platform designed to facilitate data collection and benchmarking in surgical policy learning, with a particular focus on the challenging task of suturing. The platform builds upon the AccelNet Surgical Robotics Challenge (SRC) simulator, offering a standardized pipeline for collecting and utilizing expert demonstrations, and supports various RL and IL algorithms.  A key contribution is the implementation of a hierarchical task decomposition framework, which breaks down complex surgical procedures into manageable subtasks, potentially leading to improved learning efficiency and skill transferability.  The platform also provides a diverse suite of manipulative tasks essential for suturing and establishes clear performance metrics for benchmarking different learning approaches.

The reviewers generally commend the paper for its clarity, and generally agree that this is a solid contribution to the field of surgical robotics, providing a valuable tool for researchers to develop and evaluate learning-based approaches for complex surgical tasks. The platform's compatibility with the da Vinci Surgical System, its support for hierarchical task decomposition, and its diverse suite of tasks are seen as significant strengths. The authors have also been responsive to reviewer feedback, conducting additional experiments and improving the documentation and organization of their code repository. The inclusion of expert trajectory data is also seen as a valuable resource for researchers.

There are some areas where the work could be further strengthened. The paper could more clearly articulate its specific contributions beyond the existing SRC simulator.  One reviewer notes that the current dataset may be saturated by imitation learning and TD3+BC+HER with the hierarchical system, suggesting the need for additional tasks or domains where improvement can be made. Another reviewer points out that the work does not validate the potential effectiveness of the platform in transferring learned policies to the real world, which would significantly enhance its value. Additionally, while the paper is generally well-written, there is room for improvement in clarity and conciseness in some sections. Another concern pertains to the computational cost of the simulation pipeline and the absence of wall-clock training times for the RL policies.

The authors' rebuttal addresses most of these concerns effectively. They clarify their contributions beyond SRC, provide details on computational costs and training times, and discuss ongoing efforts towards sim-to-real transfer. They also commit to improving the repository's organization and documentation. The additional experiments conducted in response to reviewer feedback, including benchmarking with dense rewards and offline RL methods, further strengthen the paper.

In conclusion, SurgicAI represents a valuable step towards advancing surgical policy learning by providing a well-structured platform for data collection, benchmarking, and algorithm development, particularly for complex tasks like suturing. The hierarchical task decomposition framework is a particularly noteworthy contribution with the potential to significantly impact the field. However, the work's overall impact is currently limited by the lack of real-world validation and the challenges associated with image-based policies. Addressing these limitations in future work will be crucial for realizing the full potential of SurgicAI and facilitating the translation of research findings into clinical practice.

Pros:
- The paper tackles the significant challenge of automating complex surgical tasks like suturing, which has broad implications for the future of robotic-assisted surgery.
- The platform offers a well-structured pipeline for data collection, training, and evaluation, promoting reproducibility and comparability of results.
- The hierarchical approach to task decomposition is a valuable contribution, enabling efficient learning and generalization across different surgical procedures.
- The platform includes a variety of manipulative operations required during suturing and provides benchmarks for assessing learned policies, facilitating further research and development.
- The platform's open-source nature and compatibility with the da Vinci Surgical System and other tools make it accessible to a wider research community.

Cons:
- The platform primarily builds upon the existing SRC simulator, with the main contributions focusing on high-level motion planning and robot learning.
- While the platform shows promising results in simulation, the paper lacks validation of the learned policies on a real robotic system, which is crucial for assessing its real-world applicability.
- The performance of image-based policies remains suboptimal, indicating a need for further research in this area.
- The repository could benefit from improved organization and documentation to enhance reproducibility.